# Deciphering Streptococcal Biofilms

**DOI:** 10.3390/microorganisms8111835

**Published:** 2020-11-21

**Authors:** Puja Yadav, Shalini Verma, Richard Bauer, Monika Kumari, Meenakshi Dua, Atul Kumar Johri, Vikas Yadav, Barbara Spellerberg

**Affiliations:** 1Department of Microbiology, Central University of Haryana, Mahendergarh, Haryana 123031, India; shaliniverma7993@gmail.com (S.V.); Kmonika480@gmail.com (M.K.); 2Institute of Medical Microbiology and Hygiene, University Hospital Ulm, 89073 Ulm, Germany; richard.bauer@uni-ulm.de; 3School of Environmental Sciences, Jawaharlal Nehru University, New Delhi 110067, India; meenakshi72@hotmail.com; 4School of Life Sciences, Jawaharlal Nehru University, New Delhi 110067, India; akjohri14@yahoo.com (A.K.J.); vikasjnu@gmail.com (V.Y.)

**Keywords:** streptococci, opportunistic pathogen, planktonic, biofilm, antibiotic therapy, quorum sensing (QS)

## Abstract

Streptococci are a diverse group of bacteria, which are mostly commensals but also cause a considerable proportion of life-threatening infections. They colonize many different host niches such as the oral cavity, the respiratory, gastrointestinal, and urogenital tract. While these host compartments impose different environmental conditions, many streptococci form biofilms on mucosal membranes facilitating their prolonged survival. In response to environmental conditions or stimuli, bacteria experience profound physiologic and metabolic changes during biofilm formation. While investigating bacterial cells under planktonic and biofilm conditions, various genes have been identified that are important for the initial step of biofilm formation. Expression patterns of these genes during the transition from planktonic to biofilm growth suggest a highly regulated and complex process. Biofilms as a bacterial survival strategy allow evasion of host immunity and protection against antibiotic therapy. However, the exact mechanisms by which biofilm-associated bacteria cause disease are poorly understood. Therefore, advanced molecular techniques are employed to identify gene(s) or protein(s) as targets for the development of antibiofilm therapeutic approaches. We review our current understanding of biofilm formation in different streptococci and how biofilm production may alter virulence-associated characteristics of these species. In addition, we have summarized the role of surface proteins especially pili proteins in biofilm formation. This review will provide an overview of strategies which may be exploited for developing novel approaches against biofilm-related streptococcal infections.

## 1. Introduction

Biofilms are surface-associated microbial communities enclosed within a self-produced matrix consisting of a single or multiple bacterial species [1]. Following the evolution of prokaryotes several billion years ago, the evolution of biofilms is considered a defense mechanism against harsh environmental conditions by providing homeostasis and protection to the involved bacterial cells [2]. Biofilms were first observed on tooth surfaces by Antoni van Leeuwenhoek in the 17th century, while the term “biofilm” was introduced into medical microbiology by Costerton in 1982 [3]. He reported biofilm formation of *S. aureus* on an infected endocardial pacemaker.

Biofilms play a prominent role in human infections. More than 80% of microbial diseases have been linked to biofilm formation. Bacterial biofilms are involved in infections of the urinary tract, the female genital tract, the bloodstream, and the upper respiratory tract [4,5,6]. Dental plaque, which contains streptococci and may eventually result in caries, is a prime example of a natural biofilm composed of a multispecies bacterial community.

In biofilms, increased resistance to or tolerance of antibiotics is a common problem. It may be intrinsic due to the microbial growth conditions or it may be caused by mutations or the exchange of antibiotic resistance genes [7,8]. As biofilms adapt to survive antibiotic treatment, infections are hard to eradicate despite proper antibiotic therapy [6,9,10].

Most streptococcal species reside as commensals on mucous membranes, while several pathogenic streptococci are responsible for life-threatening human infections. The production of biofilms is a common phenotype of commensal as well as pathogenic species. Commensal streptococci biofilms represent their natural lifeform, and in pathogenic streptococcal species, biofilms have been identified as important determinants of infectious diseases. This review will focus on specifics of streptococcal biofilms, their regulation, and the potential to interfere with biofilm production as a therapeutic approach.

## 2. Biofilm Composition

The development of these highly ordered multicellular communities is a complex multistep process [11] (Figure 1). During biofilm formation, the bacterial cells transform from planktonic life to an immotile life form [1,12], resulting in a complex community consisting of different microbial subpopulations. Biofilm formation is initiated with the adhesion of planktonic bacteria to biotic or abiotic surfaces, the development of microcolonies, and a successive production of an extracellular matrix composed of polymeric substances such as proteins, polysaccharides, and extracellular DNA [13]. Three-dimensional structures develop through maturation and finally result in the detachment of single bacterial cells [14]. Microbial cells in biofilm experience impaired diffusion of nutrients and waste products with less nutrient availability in the core of the biofilm [15]. In addition, due to increased endogenous oxidative stress within biofilms, microbial cells are subjected to spontaneous mutations [16]. Genetic variations then give rise to microbial subpopulations that are physiologically heterogeneous [17,18].

The biofilm matrix is essential for protection of bacteria from environmental stresses and consists of extracellular polymeric substances (EPS). It represents an efficient diffusion barrier, interfering with the penetration of harmful substances inside the biofilm [1]. Differential gene expression is responsible for the production of EPS, which provides cohesion of the bacterial cells and determines the structure of the biofilm. Three major components are found in the biofilm matrix in varying amounts: extracellular polysaccharides, extracellular nucleic acids, and proteins can be detected together with a high percentage of water.

### 2.1. Extracellular Polysaccharide

Microscopic evaluation of biofilms shows exopolysaccharides as elongated or branched filaments mediating the adhesion of bacteria to other bacterial cells, host cells, and abiotic surfaces [19,20]. The accumulation of polysaccharide thus acts as a molecular superglue facilitating the colonization of host surfaces. Especially exopolysaccharides of the *S. mutans* biofilm matrix have been characterized in detail and mainly consist of glucans that are synthesized by specific glycosyltransferases [21]. The extracellular polysaccharides found in biofilms differ from classic streptococcal capsular polysaccharides. Especially for pneumococci, a high amount of capsular polysaccharide appears to interfere with biofilm formation, since nonencapsulated strains demonstrate a high ability to develop in vitro biofilms, while encapsulated clinical isolates and isogenic encapsulated transformants develop less biofilm than their nonencapsulated parent strains [22,23,24,25].

### 2.2. Nucleic Acids

Extracellular DNA (eDNA) is another major component of the biofilm matrix. It is highly similar to the genomic DNA of the bacterial species present within the biofilm and is released through bacterial cell lysis [26]. It plays a role in adhesion and is essential in biofilm stabilization and maintenance. Furthermore, eDNA provides protection against antimicrobial peptides and divalent cations through chelation of these substances [26]. Upon the addition of nucleases to streptococcal biofilms, significant inhibitory and disintegrating effects on biofilm formation have been observed for *S. pneumonia, S. pyogenes,* as well as for viridans streptococci [27,28,29].

### 2.3. Extracellular Proteins

The third major component in biofilm matrix, are extracellular proteins. Extracellular proteins facilitate reorganization, degradation, and dispersal of the biofilm matrix and play a structural role in biofilms [30]. A common theme for several biofilm associated proteins of Gram-positive bacteria is their ability to form amyloids. These include BAP of *S. aureus*, EPS of Enterococci, and P1 of *S. mutans* [31,32,33]. Amyloid proteins, which assemble into insoluble fibrils, participate supportively in cell aggregation and in biofilm formation [34]. Another form of fibrillar proteins are streptococcal pili, which are highly structured cell surface appendages consisting of several different structural proteins. Their special role in the formation of biofilms will be discussed in a later section of this review. The biofilm matrix however, also contains nonfibrillar proteins like, e.g., the Glucan-binding proteins (Gbps) in *S. mutans,* which play a significant role for biofilm formation as they promote aggregation and plaque cohesion [35,36]. Furthermore, some of the proteins within the biofilm matrix are enzymes involved in the degradation of EPS and the initiation of a new biofilm lifecycle. The degradation of biopolymers also delivers energy and carbon sources to bacterial biofilm cells, especially under limited nutrient availability [37].

## 3. Biofilm Formation in Different Streptococci

Many different streptococcal species including commensal as well as pathogenic streptococci form biofilms. While biofilm formation may take place under favorable environmental conditions, many species form biofilm in adverse environmental conditions as a survival mechanism to prolong their persistence under stress.

### 3.1. Betahemolytic Group A Streptococci: Streptococcus pyogenes (GAS)

*Streptococcus pyogenes* (group A Streptococcus, GAS) is responsible for a wide range of human diseases worldwide including harmless affections of the skin as well as life-threatening toxic shock syndromes. The formation of biofilm on host cells and tissues contributes to its virulence and has been investigated in some detail [38,39]. L-glucose and D-mannose have been identified as major sugars of the GAS biofilms [40]. Further important components of the GAS biofilm matrix are pili and surface proteins of the MSCRAMM family mediating the adhesion to extracellular matrix components. A main characteristic of GAS biofilm formation is that it is very strain specific [39,41] and that the requirements for biofilm production vary among strains [9,41,42]. While certain GAS strains can form biofilm on abiotic surfaces, for other strains, a matrix protein-coated surface is required [41]. Serotype M6 and M14, e.g., are able to attach to abiotic polystyrene surfaces, while other strains (M2 and M18) need collagen types I and IV, fibrinogen, fibronectin, and laminin-coated surface to establish biofilms and some strains (M1, M12, and M49) are unable to produce biofilms at all [41].

### 3.2. Betahemolytic Group B Streptococci: Streptococcus agalactiae (GBS)

*Streptococcus agalactiae* (Group B streptococci, GBS) causes mainly infections in neonates and adult immunocompromised patients. Similar to other streptococci, GBS has the ability to develop biofilms that facilitate colonization and survival in the host [43,44,45,46]. Biofilm formation of colonizing strains obtained from asymptomatic pregnant women was increased compared to biofilm production of strains from symptomatic patients [44]. The isolation of GBS from biofilms on intrauterine devices underlines the clinical role of GBS biofilms [47]. Acidic conditions as present in the vagina promote biofilm formation in serotype III and V strains. Especially the hypervirulent serotype III clone ST-17 is an excellent biofilm former at low pH [48]. Proteins play an important part and contribute to GBS biofilm structure in these strains as treatment with proteinase K disseminates the established biofilms [48].

### 3.3. Betahemolytic Group C and G Streptococci: Streptococcus dysgalactiae subsp. equisimilis (SDSE)

*Streptococcus dysgalactiae* subsp. *equisimilis* (SDSE) is closely related to *S. pyogenes* and causes similar infections [49]. Not many studies have been conducted on biofilm formation in this species. Resembling the situation in *S. pyogenes,* the biofilm-forming ability of SDSE varies with *emm* types. Up to 46% of clinical SDSE isolates were shown to produce biofilm with stG10.0 strains being strong biofilm producers, while *emm* types stG840.0, stG6.1, and stG245.0 correlate with rather weak biofilm production [50]. In addition, extracellular DNA has been implicated in biofilm formation of SDSE [51], however, SDSE has been insufficiently investigated in regard to biofilms.

### 3.4. Biofilm in Non-Beta-Hemolytic Streptococci: Streptococcus pneumoniae

*S. pneumoniae* colonizes the human nasopharynx and is a major bacterial pathogen for upper and lower respiratory tract infections. Biofilms have been detected on host mucosal surface in clinical settings such as pneumonia, otitis media, and rhinosinusitis [29,52,53,54]. Formation of biofilm is inversely correlated with capsule formation in pneumococci [53], however, the polysaccharides present in pneumococcal biofilms remain somewhat elusive. The presence of acetyl-glucosamines and ß-linked glycopyranosyl units based on positive calcofluor white staining [55] have been implicated as components of the extracellular matrix in *S. pneumoniae*. In addition, extracellular DNA and pili proteins (RrgA, RrgB, and RrgC) have been documented [53]. Dispersion of *S. pneumonia* cells from colonizing bacterial cells in biofilms, appears to be involved in the development of invasive infections [56].

### 3.5. Biofilm in Viridans Streptococci: Streptococcus mitis Group

Streptococci of the *Streptococcus mitis* group, which include *Streptococcus oralis, Streptococcus gordonii,* and *Streptococcus sanguinis* are primarily commensals of the oral cavity and found as early colonizers of dental multispecies biofilms [57,58]. These early streptococcal colonizers are essential for the formation of stable multispecies dental biofilms [59]. Glucans are a major component of the extracellular matrix of oral multispecies biofilms and are synthesized through the glycosyltransferase (GTF) of *S. mitis* group streptococci and other oral streptococcal species. The molecular basis for glucan production in these species has been investigated in detail. *S. oralis* has a single GTF structural gene (*gtfR*), with the ability to synthesize water-soluble and -insoluble glucan using sucrose as a substrate [60]. While in *S. gordonii,* a single GTF encoding gene (*gtfG*) is positively regulated by cotranscription of *rgg,* which is present upstream of *gtfG* [61]. *S. sanguinis* also has a single GTF structural gene (*gtfP*) and synthesizes water-insoluble glucans [62]. Glucans promote the accumulation of bacteria in biofilms [63]. In oral multispecies biofilm, during later stages of biofilm development, complex intraspecies interactions may occur [64]. The cocolonization with *S. mitis* and *S. sanguinis,* e.g., enhances the biomass and the cellular metabolic activity in *C. albicans* biofilm and causes morphological changes in candida [65].

Apart from mucosal colonization, viridans streptococci can also cause invasive infections, which most often present as septicemia or endocarditis. Biofilm formation has been implicated as an important factor in infective endocarditis caused by streptococcal species from the *S. mitis* group and can be demonstrated in clinical streptococcal isolates from patients with endocarditis and sepsis [66].

### 3.6. Biofilm in Viridans Streptococci: Streptococcus anginosus Group

*Streptococcus anginosus* together with *S. constellatus* and *S. intermedius* constitutes the *S. anginosus* group (SAG). Streptococcal species of this group are found as a commensal in the oral flora, the gastrointestinal tract, the upper respiratory tract, and the urogenital tract. However, these species are also the cause of serious invasive infections including blood stream infections and abscesses. Biofilm production has been found in all of the three species [67,68,69], but little is known about the molecular determinants of *S. anginosus* biofilm formation. While some studies focus on the regulation aspect of biofilm in SAG [67,68], a detailed investigation of the biofilm matrix has not been performed. Investigations of multispecies biofilms with *S. aureus, P. aeruginosa,* and *S. anginosus,* as they occur in cystic fibrosis patients, show that *S. anginosus* is less susceptible to antibiotics within these biofilms [70].

### 3.7. Biofilm in Viridans Streptococci: Streptococcus mutans

*S. mutans* proteins involved in biofilm formation include glucan-binding proteins, collagen-binding proteins, glucosyltransferases, and the cell surface protein antigen c (PAc) [36]. Adhesive glucans produced from sucrose by glucosyltransferases (GTFs), represent an essential component of the biofilm matrix in *S. mutans* and provide the attachment of bacterial cells to surface structures [71]. In contrast to other viridans streptococci, which most often harbor a single GTF, *S. mutans* produces three types of GTFs (GtfB, GtfC, and GtfD) that are necessary to maximize the level of sucrose-dependent cellular adhesion [36]. GtfB and GtfC, which synthesize predominantly water-insoluble glucans rich in 1, 3-glucosidic linkages, are located on the cell surface and encoded by the *gtfB* and *gtfC* genes, respectively. While GtfD, which synthesizes water-soluble glucans rich in 1, 6-glucosidic linkages, has been detected in secretory proteins that is encoded by the *gtfD* gene. Each enzyme is composed of two functional domains, an amino-terminal catalytic domain (CAT), which binds and hydrolyzes the sucrose substrate, and a carboxyl-terminal glucan-binding domain (GBD), which functions as an acceptor for binding glucan, and plays a role in defining the nature of the glucan synthesized by a GTF. Multiple glucan-binding proteins (GBPs) stabilize plaque on tooth surfaces [36,72], while the cell surface protein antigen c (PAc) of *S. mutans* mediates adherence of the bacterial cells to the tooth surfaces via salivary pellicle interaction [36,73].

## 4. The Role of Pili and Streptococcal Cell Surface Proteins in Biofilm Formation (Pathogenesis and Virulence Factors)

### 4.1. Pili

Many streptococcal species harbor long proteinaceous fibril structures on their surface known as pili, which are important during initial adhesion [74]. These pili play a critical role for virulence during host–pathogen interaction and contribute to the development of biofilms. They participate in the adhesion and invasion process of the major streptococcal pathogens of humans, *S. pyogenes, S. agalactiae,* and *S. pneumoniae* [75,76,77,78].

The pili structure is built through the covalent cross-linking of two or more pilin subunits. The major subunit (backbone protein) is assembled into the pilus by a class *C* sortases that catalyzes the covalent attachment between a conserved lysyl residue of the pilin motif (WxxxVxVYPK) of one subunit and the conserved threonyl residue of the LPXTG motif of another subunit. In addition, one or more accessory subunits can be incorporated into the pilus backbone [79,80].

Ancillary protein 1 (AP-1) is attached to the tip of the main pilus component (backbone protein), and ancillary protein-2 (AP-2) anchors the pilus to the bacterial surface. Initial contact with host tissue is mediated by AP-1, which is also important for biofilm formation [81]. In addition, AP1 promotes bacterial aggregation and thus pili not only play a role in the initial adhesion leading to intimate association with host cells but also mediate the coaggregation of bacteria as an important step in biofilm formation.

In *S. pyogenes,* pili have been demonstrated to be an integral part of biofilm formation, which seems to be associated with certain M- and FCT (fibronectin-binding, collagen-binding, T-antigen-binding proteins) types [10,41,82,83,84,85,86,87]. The *S. pyogenes* T-Antigens, which have been used for serological purposes over many decades, were later shown to be pili backbone antigens [88]. The FCT region is highly variable and constitutes 9 different variants [88,89,90,91,92].

In GAS strain TW3558 (emm6), the FCT-1 pilus region (fctX operon) consisting of pilus backbone (Tee6), ancillary protein (FctX), and sortases (SrtB and SrtA) is essential for biofilm formation suggesting that both structural as well as assembly components of pili are important for adherence and biofilm formation [81,85,89] (Figure 2). While FCT type 1 strains were shown to be generally good biofilm formers, independent of media or pH-conditions, in some other FCT types, e.g., FCT-2, FCT-3, FCT-5, and FCT-6, biofilm production depends on culture conditions and low pH (Table 1) [83,86]. FCT-7 and FCT-8 are derivatives of FCT-4 and found in a very limited number of strains, so there is no study available on their biofilm-forming ability.

Streptococcal pili were first detected in *S. agalactiae* in 2005 [76]. In this species, three different pilus islands PI-1, PI-2a, and PI-2b have been identified [93,94]. GBS strains carry one or a combination of two pilus islands (PI-2A+ PI-1/PI-2B + PI-1/PI-2a+ PI-2b/PI-2a alone/PI-2b alone). These pilus islands encode classical streptococcal pili that consist of three structural proteins: PilB (the pilin backbone protein), PilA (the pilus associated adhesin at the tip of the pilus), and PilC [45]. GBS mutants with *pilB* knockouts showed a decreased ability to form biofilms and an impaired interaction with host cells compared to wild-types strains. Systemic infection with GBS lacking *pilB* resulted in enhanced clearance and reduced mortality in mouse models [95,96]. Gene deletion and complementation analysis confirmed the significant correlation between expression of type 2a pili, and the ability to form biofilm in vitro [46]. Detailed genetic studies showed that all mutations leading to a loss of pilus expression (deletion of the backbone protein PilB, deletion of sortase-encoding genes, or the deletion of ancillary protein 2 (AP-2 and PilC), which anchors the pilus to the cell wall, prevent biofilm formation. However, some of the GBS strains, not expressing pilus 2a, were also able to form biofilm, which suggests the expression of unknown factors that may compensate for the absence of type 2a pili [46]. All GBS isolates, carry one or two pilus island variant in which PI-2a was the most common among human GBS isolates, while PI-2b was the most common for GBS isolates from animal origin [46,93,94,97,98,99] (Table 2). Human GBS harboring PI-2b and animal GBS harboring PI-2a presented significantly reduced biofilm production [100]. In conclusion, strong biofilm production seems to be a common characteristic in GBS, and association of the clinical source with the pilus variant may be crucial. Different studies support the findings that GBS biofilm production, similar to the situation in *S. pyogenes*, is a lineage-specific trait in GBS [98] and is especially important for GBS colonization.

The *S. pneumoniae* pili are encoded by the *rlrA* pathogenicity islet, which carries genes for three pilin proteins a major backbone protein (RrgB), a terminal tip protein (RrgA), and a cell wall anchored base (RrgC) [101] as well as three sortases (SrtC-1, SrtC-2, and SrtC-3) (Table 2). The terminal tip protein (RrgA) mediates bacterial adherence to host ECM proteins [102,103]. The RrgB pilin backbone is composed of four immunoglobulins (Ig)-like domains (D1–D4) [104] with collagen-binding motifs. RrgC does not depend on pilus-specific sortases for attachment to the cell wall; instead, it binds the preformed pilus to the peptidoglycan by retaining the catalytic activity of SrtA [105] (Figure 2). In contrast to other streptococci, in *S. pneumoniae,* the RrgA tip protein appears to play the most important role during biofilm formation as an isogenic mutant of *rrgA* demonstrates impaired biofilm production [106], while mutations of the *rrgB* or *rrgC* genes did not influence the ability of *S. pneumonia* to form biofilm.

Pili are also present in oral streptococci belonging to the *Streptococcus mitis* group, which is closely related to *S. pneumoniae*. In *S. sanguinis*, pili are involved in colonization on saliva-coated tooth surfaces and in the human oral cavity. A pili-deficient mutant was incapable of producing a typical three-dimensional layer of biofilm and could not adhere to saliva-coated surfaces [107]. The ancillary protein at the tip of the pilus of *S. sanguinis*, which was designated PilC, binds to multiple salivary components including salivary alpha amylase. Pilus-mediated binding of *S. sanguinis* to salivary components may help this early colonizer of multispecies biofilms to attach to tooth surfaces and initiate biofilm formation in the oral cavity. In two other species of the *S. mitis* group, *Streptococcus oralis* and *S. mitis* pili were identified, which closely resemble PI-2 pili of *S. pneumoniae*, however, if there is any involvement in biofilm formation has not been investigated [108]. Surprisingly for one of the most typical streptococcal species in dental multispecies biofilms, *Streptococcus mutans*, the presence of pili has not been reported.

### 4.2. Surface Proteins and Their Role in Biofilm Formation

Apart from pili, bacterial adhesins play a direct role in the initial step of attachment to host surfaces and thus biofilm formation. For several bacterial surface protein families, their involvement in biofilm production has been demonstrated. These include the MSCRAMM family (microbial surface components recognizing adhesive matrix molecules) [109], the AgI/II family [110], the family of collagen-like proteins [111], the choline-binding proteins of *S. pneumoniae* [112], and the Bap family [31].

In *S. pyogenes*, fibronectin-binding proteins PrtF1 and PrtF2 of the MSCRAMM family are involved in biofilm formation [113,114,115], as well as IgG/fibrinogen-binding surface protein Mrp4, and the fibronectin-binding serum opacity factor of *S. pyogenes*. Another *S. pyogenes* adhesin Scl1, the streptococcal collagen-like protein 1, binds to fibronectin and has the ability to support biofilm formation as well as facilitate microcolony formation. It is conserved among all *S. pyogenes* strains investigated so far and is present in many other pathogenic streptococcal species including *S. agalactiae, S. pneumoniae*, and *S. equi* [116,117,118].

The major virulence factor of *S. pyogenes*, the anti-phagocytic M protein, which binds fibrinogen, has also been demonstrated as an important factor in biofilm formation. Biofilm production is strain dependent in *S. pyogenes* and strongly correlated with the M protein type. An association between M protein expression, surface hydrophobicity, and the ability to form biofilms for certain *emm* types has been established [119]. Furthermore, the cell wall-anchored adhesin AspA, which belongs to the antigen I/II type family, facilitate biofilm development of *S. pyogenes* on saliva-coated surfaces [120,121]. A surface protein of *S. pyogenes* with a negative influence on biofilms is the cysteine protease SpeB that promotes the dispersal of biofilms [122].

In *S. agalactiae,* the adhesive surface proteins and their role in biofilm formation have not been investigated to the same extent as in *S. pyogenes*. The role of surface proteins for biofilm production was, however, demonstrated through proteinase K treatment, which inhibited biofilm formation and induced biofilm detachment [48]. The fibrinogen-binding surface protein C (FbsC) is one of the *S. agalactiae* surface proteins investigated in more detail in regard to biofilm formation [123,124]. In addition, more recently a member of the Ag I/II family has been identified in *S. agalactiae* contributing to fibrinogen binding and involved in biofilm production [125].

The role of pneumococcal surface proteins for biofilm formation have been studied primarily in nonencapsulated strains. Among these, the choline-binding proteins (Cbps) of *S. pneumoniae*, which bind noncovalently to the phosphorocholine residues in the bacterial cell wall [126], have been demonstrated to be involved in biofilm production. Cbps are important for virulence, colonization, and adherence to host cells [112]. They include LytA (the major autolysin), LytB (a glucosaminidase involved in daughter cell separation), and LytC (a lysozyme acting as an autolysin at 30 °C), the CbpA adhesin, the PcpA putative adhesin, and PspA (pneumococcal surface protein A). A direct contribution to biofilm formation in *S. pneumoniae* has been demonstrated for LytB and LytC [27,127,128,129].

In the different viridans streptococci species especially the AgI/II and the Bap family of surface proteins play a special role in biofilm production. Common features of the Bap (biofilm-associated proteins) and Bap-related proteins are a high-molecular-weight and a core domain of tandem repeats. They are characterized as cell wall-associated proteins with amyloid behavior, involved in biofilm formation. The AgI/II proteins have been given different names according to the strains or species in which they were identified, such as antigen B, SpaP, Sr, MSL-1, and PAc from *S. mutans*; Spa A from *S. sobrinus*; PAa from *S. cricetus*; and Pas from *S. intermedius* [130,131,132].

In *S. mutans,* proteins that are required for the initiation of biofilm formation include SpaP [133]. In addition, protein antigen C (PAc) contributes to the interactions of *S. mutans* cells with fibronectin, collagen type I, and fibrinogen. Independent studies have shown that in *S. mutans* isogenic Ag I/II-deficient mutants, the initial adhesion to salivary films was reduced [134,135]. It was found that *wapA, SMU_63c,* forms amyloid like fibrils but individual mutants of *P1* (which originally identified as AgI/II or *PAc*), *wapA or 63c,* did not reduce biofilm formation in *S. mutans*. P1 forms a fibril-like structures contributing to functional amyloid formation during biofilm development. However, double and triple mutants of these genes show reduced biofilm formation [136]. As mentioned before, the glucans of the extracellular matrix play a critical role during attachment and accumulation of *S. mutans* to the host niche. Binding of *S. mutans* to these glucans is facilitated by four different cell surface-associated GBPs (glucan-binding proteins), namely, GbpA, GbpB, GbpC, and GbpD [137].

In regard to *Streptococcus parasanguis* the fimbria-associated serine-rich repeat adhesin BapA1 and Fap1 (fimbriae-associated protein 1), a high-molecular-weight glycoprotein, which is essential for assembly of fimbriae, are both critical for biofilm formation [138]. They show a similar biofilm-deficient phenotype, but function independently. Bacterial autoaggregation during biofilm formation is mediated through the N-terminal region of BapA1, by BapA1-BapA1 interactions. Deletion of the 3´ portions of *bapA1* leads to a loss of bacterial autoaggregation and reduces biofilms development [138].

In summary, numerous different cell surface proteins with the ability to bind extracellular matrix components or to mediate adhesion to host cells are also involved in biofilm production, reflecting the importance of initial attachment to host structures for the successful establishment of a streptococcal biofilm.

## 5. Regulation

The formation of biofilms is a complex multistep process that needs proper control of gene expression patterns. Thus, sensing environmental stimuli and adaptation of the gene expression is crucial and can only be realized by complex regulatory networks (Figure 3A,B).

### 5.1. GAS

The transition of a planktonic to a sessile biofilm lifestyle is associated with global changes in gene expression, which affects about 25% of all GAS genes (Table 3) [139]. During biofilm growth, classical virulence genes, typically involved in invasive disease, like the streptolysins (*sagA* and *slo*), the hyaluronic acid capsule biosynthesis (*hasA*), the M-protein, and the streptococcal pyrogenic exotoxin B (*speB*) are downregulated, whereas competence-associated genes are upregulated (Table 3) [140]. The observed downregulation of *speB* during biofilm growth plays an important role in the biogenesis of the biofilm in GAS as dispersal of the biofilm seems to be *speB* dependent and the absence of *speB* is required for biofilm growth [141]. The extracellular cysteine protease SpeB appears to be responsible for the degradation of extracellular biofilm proteins [142]. Multiple regulatory pathways are involved in the transcriptional control of *speB* [143] including seven activators (*ccpA, ropB, mga, pel, codY, sagP,* and *luxS*) and five repressors (*covRS, svr, lacD.1, nra,* and *vfr*) (Figure 3A and Table 3). Some of these regulatory pathways (CovRS, Mga, and LuxS) are indirectly linked to biofilm formation via the regulation of other effector genes.

The two-component system CovRS influences the expression of around 15% of chromosomal genes in GAS and is involved in the regulation of important virulence factors including the capsule, surface adhesins, and extracellular enzymes [144,145]. As a negative regulator of the capsule biosynthesis operon HasABC and as the capsule was shown to be involved in biofilm maturation [141], a mutation in CovRS would be expected to result in increased biofilm formation. Instead, CovS mutants of M2, M18, and M49 strains showed a reduced biofilm formation although the capsule production was increased [146]. A strain-dependent alteration of the biofilm production was observed in M6 strains, [139,146] indicating that the influence of CovRS on the biofilm regulation is serotype and even strain dependent [39]. Mga, the major stand-alone positive regulator of *emm* and *emm*-like genes, influences biofilm production besides *speB* regulation (Table 3) [147,148,149]. Neither a *mga* deletion strain nor an *emm* deletion strain retained their biofilm formation capacity potentially because of the loss of autoaggregation [139,150]. Besides *speB* regulation, the LuxS/Autoinducer-2 regulatory system is involved in the control of streptolysin S, *emm,* and capsule expression [151,152], with the latter two factors known to influence biofilm formation. Thus, the LuxS/AI-2 system could be involved in biofilm regulation in GAS, but no difference between wild-type and the corresponding *luxS*-deficient strains for the initial step of biofilm formation could be observed (Table 3) [153].

Two other quorum sensing systems, SilC and Rgg2/3, were also described as being involved in biofilm regulation in GAS as a *silC* mutant in *emm14* and *emm18* strains exhibited an altered biofilm structure [41] and the stimulation of the Rgg2/3 pathway resulted in increased biofilm biogenesis (Table 3) [154]. However, as the *sil* system is only poorly conserved and not present in the majority of clinical GAS isolates [42] and as the effector genes of the Rgg2/3 system that mediate the biofilm effect are not known yet, the contribution of these systems to biofilm regulation in GAS still needs to be further elucidated.

### 5.2. GBS

Although multiple QS (Quorum Sensing) systems are described in the regulation of adhesion and colonization in GBS, for many of these systems, no role in the regulation of biofilm was demonstrated [155,156]. An exception is CovRS, the TCS, which regulates adhesive proteins like BsaB/FbsC. *covRS*-mutant bacteria show an increased adherence to host cells and an increased formation of biofilm-like structures [123,124,157]. BceRS is also among the TCS involved in the biofilm regulation of GBS. A mutant of the response regulator *bceR* showed a reduced mortality in a murine infection model and a reduced biofilm formation potentially due to a compromised oxidative stress response [158]. Another two-component regulatory system, RgfA/C, regulates fibrinogen binding [159,160] and may thus be involved in biofilm formation (Table 3). For veterinary *S. agalactiae* strains, the quorum sensing signaling system (LuxS) has been shown to participate in biofilm formation [72].

Recently, a novel biofilm regulatory protein A (BrpA) was characterized in GBS due to the strong reduction in biofilm biomass in a *brpA*-mutant strain [161]. Although the authors investigated the transcriptome of the mutant, a conclusive reason for the loss of biofilm formation could not be identified so far as prominent surface structures like pilus type 2a and antigens I/II that were involved in GBS biofilm formation [46,125] were still present in the *brpA* mutant (Table 3).

### 5.3. S. pneumoniae

*S. pneumoniae* colonization and disease is often linked to biofilm formation. However, the regulation of the initial steps of biofilm formation and maturation are only partly understood (Table 3). Here, QS systems, the LuxS/Autoinducer-2, and the competence system are involved in the biofilm regulation in *S. pneumonia*. Although the Lux QS system seems to be important during early steps of biofilm formation [162], the competence system seems to be implicated in early and later stages dependent on the biofilm model [163,164].

The link between the competence system and biofilm formation was illusive till recently as Aggarwal and colleagues described BriC (biofilm-regulating peptide induced by competence), a peptide under transcriptional control of ComE that leads to increased biofilm biomass at later stages of the biofilm development [165]. Another small secreted peptide, VP1, was discovered in the same secretome study that identified BriC [166] and a pneumococcal mutant of VP1 formed less biofilm. The reduced biofilm formation could be restored by external VP1 addition probably due to a membrane-bound receptor recognizing VP1 at the cell surface [166].

An orthologue of the GAS Mga regulator in *S. pneumoniae*, the virulence regulator MgrA, was also described in relation to biofilm formation showing increased expression in biofilm compared to planktonic growth [23]. Additionally, *mgrA* is an important factor during nasopharyngeal colonization and pneumonia [167], two bacterial lifestyles associated with biofilm formation.

### 5.4. Viridans Streptococcus

As the etiological agent of caries, *S. mutans* is the best investigated member of the viridans streptococci concerning the regulation of biofilm formation (Table 3). Thus, multiple TCS systems are described as part of a complex regulatory network including VicRK, CiaRH, and CovRS. Reduced biofilm biomass and viable cell numbers were observed in *vic*-mutant cells [168], and a direct regulation of glycosyltransferases and glucan-binding proteins was demonstrated for VicR highlighting the role of this TCS for the adherence and early steps of biofilm growth in *S. mutans* [169,170,171]. Biofilm formation was also reduced in a *ciaH* mutant probably due to increased expression of the protease HtrA [172,173,174,175]. An orthologue of the prominent CovR regulator of GAS and GBS is encoded in *S. mutans* and was initially designated TarC [176]. Usually, CovR is the response regulator of a TCS including a histidine kinase, in *S. mutans,* however, *covR* is an orphan gene as no homologue to CovS could be detected. CovR represses the glycosyltransferases *gtfB, gtfC,* and *gtfD* and the glucan-binding protein *gbpC*, factors known to influence biofilm formation, and a *covR* mutation results in an altered biofilm architecture (Table 3) [176,177].

Two quorum sensing systems were described in relation to biofilm regulation in *S. mutans*. The bacteriocin controlling system ComDE/CSP is involved in the programmed cell death and persister cell formation during biofilm growth, and the LuxS/Autoinducer-2 system participates in the regulation of the early steps of biofilm formation [178,179,180,181].

A central role in the regulation of biofilm formation in *S. mutans* is realized by BrpA, an orthologue of GBS BrpA. The expression of more than 200 genes is altered in a *brpA* mutant including genes involved in stress tolerance, adherence, and cell wall biogenesis resulting in a profound biofilm-formation defect [182,183,184]. Another regulatory protein the peptidyl-prolyl isomerase (RopA) plays a substantial role in biofilm formation with or without saliva and sucrose [185]. Transcriptional regulator BrpT negatively regulates the *gtfP* expression as deletion of *brpT* promotes the expression of *gtfP* and displays increased biofilm formation ability.

A link between exopolysaccharide synthesis and biofilm formation was demonstrated by the characterization of *rnc*. A-mutant strain showed an altered biofilm formation and reduced bacterial adhesion. Additionally, *rnc* affects the expression of the TCS VicRK, which itself is involved in biofilm regulation [186,187]. Two additional regulators were reported to be involved in biofilm control in *S. mutans*. The StsR protein, a GntR transcription factor, seems to positively regulate biofilm formation and EPS production, although the effect was not to pronounced and the mutant of *stsR* showed a growth delay, which could be responsible for the observed phenotypes [188]. The other factor NagR was demonstrated to regulate the expression of *glmS* and *nagAB,* which are known to be involved in biofilm formation [189,190]. However, a *nagR*-mutant strain was so far not tested for biofilm formation; thus, it remains elusive if the regulation of NagR directly affects biofilm growth.

Only little is known about other members of the viridans streptococci concerning biofilm regulation (Table 3). The LuxS/Autoinducer-2 QS system is described in *S. anginosus* and *S. gordonii,* with the difference that a *luxS* mutant in *S. anginosus* seems to be almost deficient in biofilm formation, whereas a *luxS* mutant in *S. gordonii* and in *S. suis* formed an altered microcolony structure within the early biofilm [68,191]. Additionally, CcpA, the catabolite control protein A, was described as important for the biofilm formation of *S. gordonii and S. mutans*, as a *ccpA*-mutant strain showed a reduced biofilm biomass under various growth conditions [192]. In *S. sanguinis,* the regulation of the glycosyltransferase *gtfP* was described in two independent mutant strains, *ciaR* and *brpT,* both demonstrating positive association of *gtfP* expression and biofilm formation [62,193]. In *S. gordonii*, TCS system named “BfrAB,” which encodes two ABC transporters and a CAAX amino-terminal protease family protein, promotes biofilm development [72].

Sugar metabolism enzyme was identified in *S. suis,* and the gene involve in the process is said to be as pyruvate dehydrogenase (*pdh*), which acts as temperature stress, salt stress, and oxidative stress controller and involves in adhesion, biofilm formation, and antiphagocytic activity [194].

In *S. sanguinis,* SptRSs and SptSSs are the TCS systems that coordinate cell wall homeostasis and are involved in H_2_O_2_ production and competence [195]. Another quorum sensing system is homoserine kinase (*thrB*), which is involved in threonine biosynthesis. Metal (Zn)-binding permease (*adcA*) shows implication in biofilm formation by *S. gordonii Challis* and *S. sanguinis* [196]. There are three genes, which are identified and found to be responsible for de novo pyrimidine biosynthesis, and these are orotate phospho-ribosyl-transferase (*pyrE*), phospho-ribosyl-formyl-glycin-amidine synthase (*purL*), and adenylosuccinate lyase (*purB*) (Figure 3B and Table 3) [196].

## 6. Therapeutic Antibiofilm Approaches

The treatment of bacteria growing in biofilms is challenging due to the increased tolerance towards antibiotics and their protection against the host immune system. Identification of agents that interfere with biofilm formation may increase the effectiveness of antimicrobials and allow access of host defenses, which may then resolve the infection. To prevent biofilm formation, innovative approaches aim at specific therapeutic targets. These may include biofilm regulation, degradation of the extracellular matrix, and the targeted delivery of antimicrobial peptides and other antibiofilm agents.

### 6.1. Quorum Sensing Systems

Biofilm formation shows a close biological connection with the quorum sensing (QS) systems found in many streptococci. Several different ways can be used to interrupt QS-associated cell communication system including: competitive inhibition of signaling molecules, compounds with signaling molecules binding capacity, and degradation of the chemical structures of signaling molecules. Interestingly signaling molecules like the competence-stimulating peptide of *S. mutans* may at higher concentrations cause bacterial self-destruction and thus interfere with biofilms [218]. KBI-3221, an analog of the competence-stimulating peptides of various Streptococci, reduces biofilm formation in several species [219]. In another study, the QS mechanism interfering compound 5-hydroxymethylfurfural was shown to inhibit biofilm formation in different streptococcal species [220]. These studies highlight that an interference with quorum sensing regulation appears a promising antibiofilm strategy, which may also represent a more general approach to interfere with host microbe interactions [221].

### 6.2. Target Extracellular Polysaccharide (EPS) Matrix

Biofilm matrix components, such as polysaccharides, proteins, and eDNA, are crucial for providing resistance against antimicrobials, the immune system, and environmental stressors [222,223]. The inhibition of biofilm formation can thus be achieved by matrix degrading enzymes that can induce biofilm dispersion, biofilm detachment, and reduction in cell aggregation [224]. While different types of compounds have been shown to degrade EPS in *E. coli*, *P. aeruginosa,* and staphylococci [225,226,227], a few studies exist on streptococci. DNase I as well as different proteases have been shown to degrade biofilm matrices in streptococci [29,228,229], despite potential toxicity to the host they may be useful in developing novel therapeutic approaches [230]. In 2015, Ren et al. reported a more specific approach to target the EPS of oral multispecies biofilm [231]. They identified a compound termed 5H6 as an inhibitor of glucosyltransferases, a key contributor to the EPS of *S. mutans,* and they could show the effect of 5H6 on preventing caries development in an in vivo rat model.

### 6.3. Antimicrobial Peptides (AMPs)

Most antimicrobial peptides are small cationic and amphipathic peptides that are part of the host innate immune system. AMPs have been shown to interfere with various stages of biofilm formation independent from their ability to kill bacterial cells [232]. Some of the most prominent examples of AMPs showing biofilm preventive activity include the human cathelicidin peptide LL-37 (derivative of mucosal epithelial cells and several cells of the immune system), Lactoferrin, Oritavancin (semisynthetic lipoglycopeptide), DJK-5, and DJK-6 (synthetic analog of active antibiofilm peptides), DD13-RIP (chimeric compound), and (IDR-)1018 (bactenecin derivative) [233,234]. In regard to streptococci Chrysophsin-1, IDR-1018 and pleurocidin have been shown to exhibit antibiofilm activity [218,235,236,237]. The AMP C16G2 may even selectively remove *S. mutans* from multispecies oral biofilms and shift the microbial community towards beneficial streptococcal species [238]. An attractive approach is to strengthen the antibiofilm activity of AMPs by targeted delivery. Specificity, e.g., can be increased by conjugation to a pathogen-specific siderophore, packaged within a phage delivery system [239]. Furthermore the combination of an AMP with conventional antibiotics can strengthen the antimicrobial activity, at sub-MBIC (minimal biofilm inhibition concentration) levels, IDR-1018, e.g., causes a 64-fold decrease in the respective MBIC of several antibiotics [239]. In summary, coadministration of AMPs with antibiotics or other antimicrobial strategies may offer an effective way to combat streptococcal biofilms.

### 6.4. Bacteriocins

Bacteriocins are antimicrobial peptides, which are produced by bacteria to kill closely related bacteria. While they may have similar antibiofilm properties as eukaryotic AMPs, these have not been investigated to the same extent. Nisin has been shown to interfere with biofilms in several bacterial species including enterococci [240,241], there are, however, conflicting reports about the activity of nisin on *S. mutans* biofilms [242,243]. In summary the potential antibiofilm properties of bacteriocins have not been sufficiently explored for streptococci.

### 6.5. Nanodrug Delivery System

Nanoparticles are an emerging technology used for drug delivery and selective targeting of pathogenic bacteria with the ability to penetrate biofilms [230]. Liposomes, e.g., can carry more than one drug by coencapsulation and can also be functionalized by linking biomolecules such as peptides or polymers to increase target specificity and to provide a triggered release [244]. Inorganic nanoparticles such as iron oxide (Fe_3_O_4_) with a peroxidase-like function catalyzes hydrogen peroxide (H_2_O_2_) showing potent effects against *S. mutans* oral biofilms in vivo [245]. Several other studies using metallic nanoparticles underline their potential as antibiofilm strategies [246,247,248,249]. Overall, nanoparticles offer a promising therapeutic platform for the development of new and effective biofilm-targeting approaches.

### 6.6. Surfactants, Amino-Acids, Metal Chelators, and Various Enzymes

Surfactants show antibiofilm activity by interacting with various cellular components, such as proteins and lipids, reducing microbial cell growth and viability [250,251,252]. Sodium dodecyl sulfate (SDS) promotes biofilm dispersion by causing cavity formation within biofilms [251]. Several other surfactants, such as surfactin, rhamnolipids, Tween 20, cetyltrimethylammonium bromide (CTAB), and Triton X-100, have also been reported to cause biofilm disruption [253,254]. The disintegration of biofilms can also be achieved by amino acids (D-cysteine (Cys), D- or L-aspartic acid (Asp) and D-or L-glutamic acid (Glu), which have been reported for various bacterial species including streptococci [243,255].

### 6.7. Phages Therapy

Bacteriophages may also be used for their antibiofilm properties. They have the advantage of being specific for bacterial strains and of having a rapid, targeted action, reducing the development of resistance [256]. Even for resistant bacteria, phages are often bacteriocidal. The successful application of bacteriophages or bacteriophage-encoded lysins was shown for *S. pyogenes* and *S. suis* [257,258,259]. More recently, a novel *S. mutans* phage was isolated showing antibiofilm activity [260] and the use of a bacteriophage-encoded lysin against *S. mutans* biofilms could be demonstrated [261].

## 7. Conclusions and Future Prospective

Biofilms are often essential in the development of streptococcal infections and represent an important clinical challenge affecting morbidity and mortality. They provide defense against antibiotics and protection from the immune system, therefore treatment with antimicrobial drugs alone is difficult. Addressing this situation requires a better understanding of the underlying molecular mechanisms. In this review, we discuss the role of virulence factors, pili, and surface proteins and regulators associated with streptococcal biofilms. Targeting different aspects of biofilm production, promising antibiofilm approaches can be envisioned. Various in vitro studies show that streptococcal biofilms can be attacked by addressing quorum sensing systems, bacteriocins, EPSs, AMPs, nanodrugs, enzymes, surfactants, and phages. Thus, biofilms that represent a crucial part in the development *Streptococcal* infections can be a key target for novel therapeutic strategies.

## Figures and Tables

**Figure 1 microorganisms-08-01835-f001:**
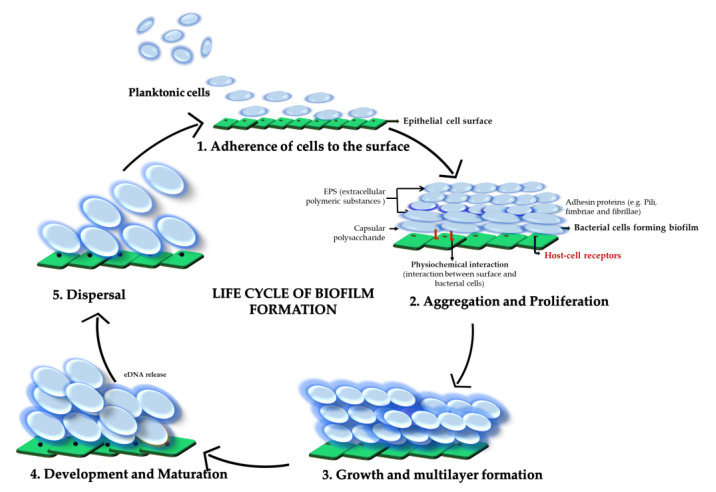
Schematic diagram representing the life cycle of biofilm formation in Streptococci. The diagram shows the transition of planktonic cells to sessile cells by undergoing different stages of biofilm formation and repeating the cycle by the conversion of sessile cells to the planktonic state again.

**Figure 2 microorganisms-08-01835-f002:**
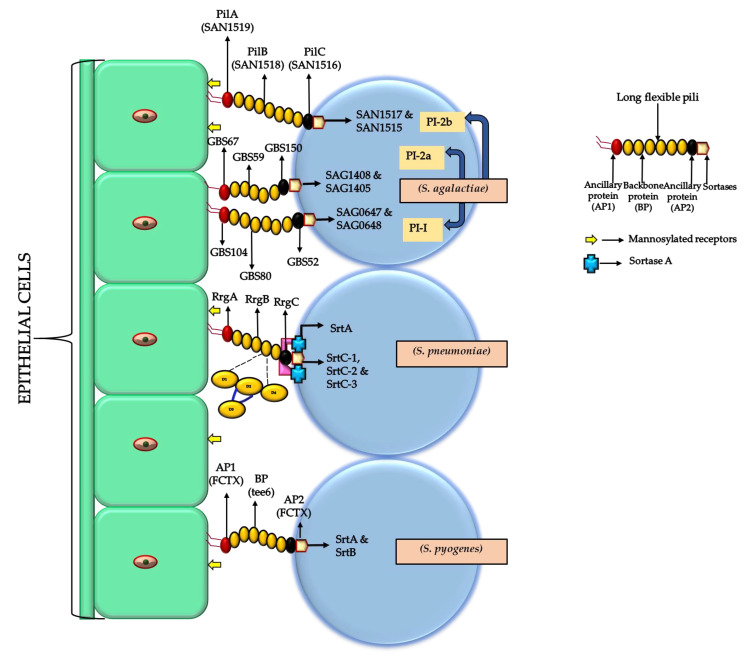
Schematic diagram of general pilus architecture in different streptococci. The blue spheres represent the different bacterial strains, and the green color boxes represent host epithelial cells. The long pili shown in the diagram are formed by three major subunits ancillary protein AP1 (tip protein at the distal end) that attaches to the bacterial surface, backbone protein BP, and ancillary protein AP2, which are assembled by sortases that facilitate adherence with host epithelial cells. AP2 anchors the pilus to the bacterial surface. In *S. agalactiae*, there are three pilus islands designated as PI-1, PI-2a, and PI-2b. Each pilus island has major protein referred as PI-1 (include GBS104 as AP1, GBS80 as BP, and GBS52 as AP2 with two sortases SAG0647 and SAG0648), in PI-2a (GBS67 as AP1, GBS59 as BP, and GBS150 as AP2 with two sortases SAG1408 and SAG1405), and PI-2b (SAN1519 as AP1, SAN1518 as BP, and SAN1516 as AP2 with two sortases SAN1517 and SAN1515). In *S. pneumoniae*, pilus subunits are AP1 (RrgA), BP (RrgB), and AP2 (RrgC) with sortases designated as SrtC-1, SrtC-2, and SrtC-3. Here, RrgC does not depend on pilus-specific sortases for adherence to the cell wall; instead, it binds the preformed pilus to the peptidoglycan by retaining the catalytic activity of SrtA (blue box). RrgB is composed of 4 domains D1 at N-terminus, D2 and D3-positioned laterally, D3 connected to D2 through a loop (dark blue color), and D4 at the C-terminus. In *S. pyogenes*, the three subunits of pilus include pilus BP *(tee6),* AP1 and AP2 *(fctX),* and sortases *(srtB and srtA)*. The monosyl receptors are present on the surface of epithelial cells and respond to the infection (arrow in yellow color).

**Figure 3 microorganisms-08-01835-f003:**
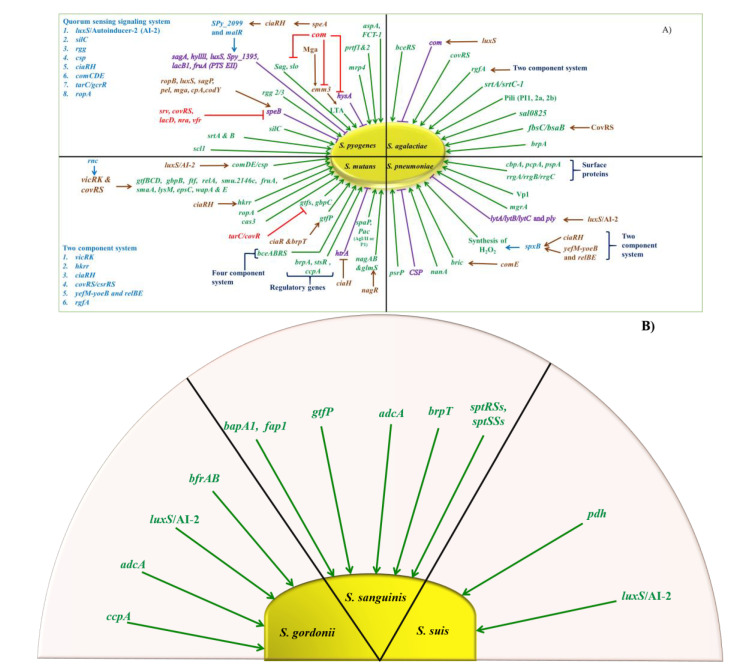
(**A**) Regulation of biofilm in different streptococcal species at the genetic level. The figure describes the gene regulation system in *S. pyogenes, S, mutans, S. agalactiae,* and *S. pneumoniae* for direct effect on biofilm inhibition (shown in purple), induction (shown in green), negative regulators (shown in red), and positive regulators (shown in brown and blue). (**B**) Involvement of virulence genes in the regulation of biofilm in different species of the *Streptococcus mitis* group and *S. suis*. The represented genes of *S. gordonii, S. sanguinis,* and *S. suis* are involved in the positive regulation of biofilm formation and thus directly involved in increasing the biomass of forccpamed biofilm.

**Table 1 microorganisms-08-01835-t001:** Representation of *S. pyogenes* pilus structure encoding in the Fibronectin-Binding, Collagen-Binding, T-Antigen (FCT) genomic region, and its effect on biofilm-forming capacity.

FCT (Fibronectin-Binding, Collagen-Binding, T-Antigen) Type *Encoding pili*	Biofilm Phenotype
FCT type 1	Strong biofilm, independent of media or pH
FCT type 9	Poor biofilm, under all tested condition
FCT-2, FCT-3, FCT-5, and FCT-6	Biofilm phenotype dependent upon on culture conditions and is triggered by low pH
FCT type 4	Inhomogeneous response to environmental conditions with respect to biofilm formation.

**Table 2 microorganisms-08-01835-t002:** Genomic distribution of different pilus island in different *Streptococci*.

Streptococci Species	Pilus Operon	Backbone Protein	Ancillary Protein-1 (Adhesin)	Ancillary Protein-2	Gene Encoding for Sortase Enzyme
*S. agalactiae*	PI-1	* GBS80 (* mandatory)	GBS 104	GBS 52	SAG 0647 and SAG 0648
*S. agalactiae*	PI-2a	GBS 59	GBS 67	GBS 150	SAG 1408 and SAG 1405
*S. agalactiae*	PI-2b	SAN 1518	SAN 1519	SAN 1516	SAN1517 and SAN 1515
*S. pyogenes*	*fctX* operon	Tee6	fctX	fctX	SrtB and SrtA
*S. pneumoniae*		RrgB	RrgA	RrgC	SrtC-1, SrtC-2, and SrtC-3

**Table 3 microorganisms-08-01835-t003:** List of different genes present in *Streptococcal* species that are involved in biofilm formation and act as adhesins, virulence factors, or regulators.

Virulence Factors	Genes	Function	Streptococci Species	Reference
Quorum sensing system	*luxS*	Key regulator of early biofilm formation	*S. gordonii*	[61]
		Involvement in virulence, competence, biofilm formation, acid and oxidative stress tolerance, and carbohydrate metabolism	*S. agalactiae*	[72]
		Regulation of *lytA* and early biofilm formation	*S. pneumoniae*	[162,197]
		Regulatory role in biofilm formation	*S. mutans*	[178]
		Key regulator of early biofilm formation	*S. suis*	[198,199]
	Streptococcal invasion locus (*silC)*	Regulator affecting biofilm architecture and density	*S. pyogenes*	[41]
	Regulatory gene of glucosyltransferase (*rgg*)	Universal streptococcal regulator involved in intraspecies communication, increased biogenesis of biofilms by *rgg2 rgg3*	*S. pyogenes*	[142]
	Competence-stimulating peptide (*CSP*)	Competence development, involved in later stages of biofilm production	*S. pneumoniae*	[23,200]
	*comCDE*	Regulation of competence through production of competence-stimulating peptide (CSP)	*S. mutans*	[178,181]
Two-component signaling system	*bfrAB*	Regulator involved in the maturation of multispecies biofilms	*S. gordonii*	[72]
	*bceRS*	Control of oxidative stress response and biofilm production	*S. agalactiae*	[158]
	Histidine kinase (*ciaH*)	Regulatory role in biofilm formation, acid tolerance, and genetic competence	*S. mutans*	[172]
	vicR/K system (*vicK*)	Modulates the expression of several genes such as *gtfBCD, gbpB, ftf, wapE, smaA, SMU.2146c, lysM,* and *epsC* that affect the synthesis of EPS and biofilm formation	*S. mutans*	[169,201,202]
	*ciaR/H*	Control of the competence operon	*S. pneumoniae*	[203]
	*yefM-yoeB* and *relBE*	Control of resistance towards oxidative stress and involvement in biofilm formation	*S. pneumoniae*	[204]
	*rgfA*	Control of adherence to fibrinogen	*S. agalactiae*	[159,160]
	*covR/S*	Major virulence and adherence regulator		[123,124,157]
	Histidine kinase (*hk11*) and response regulator (*rr11*)	Control of biofilm formation and acid resistance	*S. mutans*	[205]
	*sptRSs* and *sptSSs*	Coordination of cell wall homeostasis, involved in H_2_O_2_ production, and competence	*S. sanguinis*	[195]
Four-component system	*bceA, bceB, bceR*, or *bceS*	Regulation of sensitivity towards antimicrobial peptides and requirement for biofilm formation	*S. mutans*	[206]
CRISPR/Cas systems	*cas3* gene	Bacterial immunity, effect on biofilm formation, and fluoride sensitivity	*S. mutans*	[202]
Extracellular enzyme	Glucosyltransferases (*gtfB*, *gtfC,* and *gtfD*) and fructosyltransferases (*ftfs*)	Carbohydrate metabolism for the generation of exopolysaccharide	*S. mutans*	[207]
Sugar metabolism enzyme	Pyruvate dehydrogenase (*pdh*)	Control of environmental stress and promotion of biofilm formation	*S. suis*	[194]
Glucan binding	Glucan-binding protein (*gbpA, gbpB,* and *gbpC*)	Adhesion and promotion of biofilm formation	*S. mutans*	[208,209]
Amyloid proteins	Wall-associated protein (*wapA* and *wapE*)	Production of extracellular matrix	*S. mutans*	[136]
Regulatory proteins	Biofilm regulatory protein (*brpA*)	Regulation of acid and oxidative stress tolerance and biofilm formation	*S. mutans*	[182,205,210]
		Virulence regulator stress tolerance	*S. agalactiae*	[211]
	Sugar Transporter Systems Regulator (*stsR*)	Formation of biofilm and production of extracellular polysaccharides (EPS) at early stage	*S. mutans*	[188]
	Catabolite control protein (*ccpA*)	Global transcriptional regulator of carbon catabolite repression, involvement in biofilm formation	*S. mutans*	[210]
Surface protease	Serine protease (*htrA*)	Processing and maturation of extracellular proteins including surface associated glycolytic enzymes (GbpB, GtfB, and FTF) contributing to biofilm formation	*S. mutans*	[174]
Surface-associated proteins	Fimbria-associated serine-rich repeat adhesin (*fap1*) and (*bapA1*)	Adhesins with important role in biofilm initiation	*S. sanguinis*	[212]
	Choline-binding protein adhesin (*cbpA*), putative adhesin (*pcpA*), and pneumococcal surface protein A (*pspA*)	Adhesins binding to the teichoic acids of the cell wall, involvement in immune evasion, and promotion of biofilm formation	*S. pneumoniae*	[27,53]
	Pneumococcal serine-rich repeat protein (*psrP*)	Adhesion to host cells and mature biofilm formation	*S. pneumoniae*	[213]
	Pyruvate oxidase (*spxB*)	Responsible for the synthesis of H_2_O_2_		[53,214]
Pili/fimbriae	Genomic island (PI-1, -2a, -2b). All islands contain 3 genes encoding pilus component	Pilus assembly and creation of biofilms	*S. agalactiae*	[93,94,95]
	*rrgA, rrgB, and rrgC*	Pilus subunits and involvement in biofilm formation	*S. pneumoniae*	[106]
	FCT-1 region (*fctX, srtB, and tee6*)	Pili and biofilm formation	*S. pyogenes*	[84]
Adhesin	Bacterial surface adhesin of GBS (BsaB) (*sal0825*)	Attachment of GBS to epithelial cells, extracellular matrix and promotion of biofilm production	*S. agalactiae*	[124]
	Fibrinogen-binding protein (*fbsC*)	Fibrinogen binding, promotion of invasion of epithelial and endothelial barriers, biofilm formation	*S. agalactiae*	[123]
Antigen	Neuraminidase (*nanA*)	Release of sialic acid residues, modification of immune defense proteins, promotion of colonization and biofilm formation	*S. pneumoniae*	[215]
Autolysin	*lytA* (amidase), *lytB* (glucosaminidase), and *lytC* (lysozyme)	Cell separation, autolysis and promotion of biofilm dispersion	*S. pneumoniae*	[27,216]
M-protein	*emm*	Key virulence factor, antiphagocytic, immune evasion, adhesin, and contribution to biofilm formation	*S. pyogenes*	[38,119,142]
Hyaluronic acid capsule	Hyaluronate synthase (*hasA*)	Escape of phagocytosis involvement in biofilm maturation	*S. pyogenes*	[38,139]
Sorting signal	Sortase A (*srtA*), sortase C (*srtC-1)*	Pilus polymerization and cell wall attachment	*S. agalactiae*	[43,46]
	Sortase (*srt A* and *srtA*)	Pili assemblance and biofilm production	*S. pyogenes*	[39]
Transcriptional regulator	Streptococcal regulator of virulence (*srv*)	Transcriptional regulator of virulence and contribution to biofilm dispersal by degrading SpeB	*S. pyogenes*	[122,142]
Streptococcal antigen I/II (AglI/II) family polypeptides	Group A Streptococcus protein A (*aspA*)	Adhesion to human salivary glycoproteins and facilitation of colonization to develop biofilm	*S. pyogenes*	[38,39]
Collagen-like protein	Streptococcal collagen-like gene-1 (*scl-1*)	Cell surface adhesin	*S. pyogenes*	[142]
MSCRAMM family proteins	Fibronectin-binding protein F (*prtF1* and *prtF2*) and *mrp4*	Adherence to host epithelial cells	*S. pyogenes*	[38,39,111]
Exotoxin	*(speA)*	Superantigen involved in the dispersal of biofilm	*S. pyogenes*	[122,217]
	Cysteine protease *(speB)*	Cleavage of streptococcal cell surface virulence factors such as M protein, protein F, and C5a peptidase. Dispersal of biofilm		

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
