# Peer review of "Deciphering Streptococcal Biofilms"

_microorganisms, 2020, doi:10.3390/microorganisms8111835_

Round 1

Reviewer 1 Report

The review manuscript of Verma et al is a nice piece of work and collates important data on the building blocks of biofilm formation in Streptococcal species and also briefly outlies potential and current therapeutic strategies to combat related biofilm growth. The review brings together past and recent research and gives an overview of molecules used by Streptococcal species in biofilms. This is a much improved read a much easier to follow since modifications have been made. The review is interesting but there are still a few points that need to be addressed.

Major

  • Section 1 – Biofilm composition – there has been no mention of the role of lipids in Strep biofilm formation. The authors should at least mention the role of such things as OMVs in biofilm development and mixed community growth.

  • Line 101: Proteins also provide a structural role in biofilms.

  • There is a general problem through the manuscript where tables and figures are either not references at all in the main text or where they need to be referenced in more than one place – e.g. every time data from the table is referred to it should be referenced. Please correct throughout.

  • 1: The figure needs modifying so that you can easily distinguish cells from other components like capsule and EPS.

  • Line 178: “Whereas …” – the authors already mention that sucrose is utilized over glucose above so why is it important to repeat it here? I also do not understand the sentence in lines 180-182. Please clarify this so I can be understood.

  • There is a general problem through the manuscript where genes are written but need to be put in italics – please correct throughout.

  • Figure 2: There are major errors with this schematic, or it is not clear and needs fixing. This has not been explained in the legend, but I assume the blue spheres are bacteria, and the green blocks are epithelial cells. Why it the base of the pilus (e.g. sortase) attached to the epithelial cells? It looks like the pilus has been drawn the wrong way around. It should be protruding from the bacteria and the tip interacting with the host cells. Please correct and draw in such a way that this is clearer in general. Also use the legend to describe the figure.

  • Section 3.2: the title of the section includes “.. their role in biofilm formation.” Therefore, make sure here that all examples that are given have a role associated with them. In general, this has been done but for example in line 324, “… promotes biofilm formation …” is not informative. Either be more precise or if this knowledge is not known, say that.

  • Throughout the manuscript, some paragraphs are very short – sometimes just one sentence. Please go through the document and use proper paragraph structuring.

  • Figure 3. Why does A and B have a separate figure legend? If they are separate figure, they should have separate figure numbers. Also, the resolution of Fig 3A is too low and cannot be read.

  • Tables 3 to 7: these tables are just long list of data and hard to interpret. Instead they should be combined into a single table and presented in such a way that different virulence factor types can be compared between different species. E.g. have a single heading for QS and then collate all the same genes in the next column and then the strep species in the next column etc. This way it will be easier to compare systems from different Strep species and will make the review more compact document.

  • Line 564: I do not understand what “more or less specific” means – please clarify.

  • Line 613: which types of amino acids – provide details

  • I do not understand the sentence on lines 625 to 626. Please reword this so it is clearer.

Minor

  • Line 40: “Biofilm was first …” should be “Biofilms were first …”
  • Line 48: “In biofilm …” should be “In biofilms …”
  • Line 50: “As biofilm adapt …” should be “As biofilms adapt …”
  • Line 74: “… harmful substance inside …” should be “… harmful substances inside …”
  • Line 176: “… using sucrose as substrate.” should be “… using sucrose as a substrate.”
  • Line 177: “… single encoding gene ...” should be “… a single encoding gene ...”
  • Line 248: the text is a different font size
  • Table 1: “Ph” should be “pH”
  • Line 364: “… did not reduces …” should be “… did not reduce …”
  • Fig 3A legend: “… species at genetic level.” Should be “… species at the genetic level.”
  • Line 546: “Quorum sensing system” should be “Quorum sensing systems”
  • Line 547: “Quorum sensing (QS) system” should be “Quorum sensing (QS) systems”
  • Line 551: remove the word “itself”
  • Line 555: “compound,5-hydroxy…” should be “compound 5-hydroxy…”
  • Line 559: “… polysaccharide …” should be “… polymeric …”
  • Line 613: “… biofilm …” should be “… biofilms …”

Author Response

Major          

  • Section 1 – Biofilm composition – there has been no mention of the role of lipids in Strep biofilm formation. The authors should at least mention the role of such things as OMVs in biofilm development and mixed community growth.

According to the reviewer’s comment, we searched for the lipids role in biofilm. But we were not able to find any relative information regarding the role of lipids in Streptococcal biofilm formation. However, we find out the involvement of the lipids in biofilm formation in other bacteria.

  • Line 101: Proteins also provide a structural role in biofilms.

We modified this line as follows:

Extracellular proteins facilitate reorganization, degradation, dispersal of the biofilm matrix, and play a structural role in biofilms [30].

Old version:

Extracellular proteins facilitate reorganization, degradation, and dispersal of the biofilm matrix [30]

  • There is a general problem through the manuscript where tables and figures are either not references at all in the main text or where they need to be referenced in more than one place – e.g. every time data from the table is referred to it should be referenced. Please correct throughout.

We added the reference of tables and figures in the main text where it was required to be mentioned.

  • 1: The figure needs modifying so that you can easily distinguish cells from other components like capsule and EPS.

We have modified Fig.1 as suggested by the reviewer.

  • Line 178: “Whereas …” – the authors already mention that sucrose is utilized over glucose above so why is it important to repeat it here? I also do not understand the sentence in lines 180-182. Please clarify this so I can be understood.

We have modified line 178 as suggested by reviewer as follows:

  1. sanguinis also has a single GTF structural gene (gtfP) and synthesizes water insoluble glucans

Old version:

Whereas, S. sanguinis utilizes sucrose not glucose for polysaccharide production. It has a single Gtf structural gene (gtfP) and synthesize water insoluble glucans [62].

We have modified the line 180- 182 as follows;

In oral multispecies biofilm, during later stages of biofilm development, complex intraspecies interactions may occur

Old version:

Oral multispecies biofilms which may include S. mutans as well as Candida albicans in later stages of development present with complex intraspecies interactions

  • There is a general problem through the manuscript where genes are written but need to be put in italics – please correct throughout.

We checked the whole document and corrected wherever the genes names were not mentioned in italics.

  • Figure 2: There are major errors with this schematic, or it is not clear and needs fixing. This has not been explained in the legend, but I assume the blue spheres are bacteria, and the green blocks are epithelial cells. Why it the base of the pilus (e.g. sortase) attached to the epithelial cells? It looks like the pilus has been drawn the wrong way around. It should be protruding from the bacteria and the tip interacting with the host cells. Please correct and draw in such a way that this is clearer in general. Also use the legend to describe the figure.

The Fig. 2 is now modified and also the legend contains some changes, so that the information about epithelial cells and bacterial cells could be made clearer.

  • Section 3.2: the title of the section includes “.. their role in biofilm formation.” Therefore, make sure here that all examples that are given have a role associated with them. In general, this has been done but for example in line 324, “… promotes biofilm formation …” is not informative. Either be more precise or if this knowledge is not known, say that.

As per reviewer’s suggestion, we have changed the line (324-326, new numbering) as follows:

Another S. pyogenes adhesin Scl1, the streptococcal collagen-like protein 1, binds to fibronectin, and has the ability to support biofilm formation as well as facilitate microcolony formation.

Old version:

Another S. pyogenes adhesin Scl1, the streptococcal collagen-like protein 1, binds to fibronectin, promotes biofilm formation and seems to facilitate microcolony formation 

  • Throughout the manuscript, some paragraphs are very short – sometimes just one sentence. Please go through the document and use proper paragraph structuring.

We have considered the reviewer’s suggestion to go through the document and accordingly we rearranged the paragraph structuring again.

  • Figure 3. Why does A and B have a separate figure legend? If they are separate figure, they should have separate figure numbers. Also, the resolution of Fig 3A is too low and cannot be read.

In Figure 3, we now made the combined legend for A and B. Also, the figure is corrected with better resolution.

  • Tables 3 to 7: these tables are just long list of data and hard to interpret. Instead they should be combined into a single table and presented in such a way that different virulence factor types can be compared between different species. E.g. have a single heading for QS and then collate all the same genes in the next column and then the strep species in the next column etc. This way it will be easier to compare systems from different Strep species and will make the review more compact document.

Reviewer’s suggestion on compiling all the tables 3- 7 is considered as a great suggestion. Therefore, we compiled all the tables in Table3. List of different genes present in the Streptococcal species that involve in biofilm formation and act as virulence factors.

  • Line 564: I do not understand what “more or less specific” means – please clarify.

Here, in line 564 (552-553, new numbering), we modified it to “different types of compounds” as follows.

 While different types of compounds have been shown to degrade EPS in E. coli, P. aeruginosa and staphylococci [233–235], few studies exist on streptococci.

Old Version

While different more or less specific compounds have been shown to degrade EPS in E. coli, P. aeruginosa and staphylococci [236–238], few studies exist on streptococci.

  • Line 613: which types of amino acids – provide details

The details for the types of amino acids are now mentioned in line 601 (new numbering).

  • I do not understand the sentence on lines 625 to 626. Please reword this so it is clearer.

The sentence on line 625- 626 (614-616, new numbering) has been modified as per the reviewer’s suggestion as follow:

They provide defense against antibiotics and protection from the immune system, therefore treatment with antimicrobial drugs alone is difficult.

Old version:

Due to the protection biofilms provides towards antibiotics and the immune system treatment with antimicrobial drug alone is difficult.

Minor

  • Line 40: “Biofilm was first …” should be “Biofilms were first …”
  • Line 48: “In biofilm …” should be “In biofilms …”
  • Line 50: “As biofilm adapt …” should be “As biofilms adapt …”
  • Line 74: “… harmful substance inside …” should be “… harmful substances inside …”
  • Line 176: “… using sucrose as substrate.” should be “… using sucrose as a substrate.”
  • Line 177: “… single encoding gene ...” should be “… a single encoding gene ...”
  • Line 248: the text is a different font size
  • Table 1: “Ph” should be “pH”
  • Line 364: “… did not reduces …” should be “… did not reduce …”
  • Fig 3A legend: “… species at genetic level.” Should be “… species at the genetic level.”
  • Line 546: “Quorum sensing system” should be “Quorum sensing systems”
  • Line 547: “Quorum sensing (QS) system” should be “Quorum sensing (QS) systems”
  • Line 551: remove the word “itself”
  • Line 555: “compound,5-hydroxy…” should be “compound 5-hydroxy…”
  • Line 559: “… polysaccharide …” should be “… polymeric …”
  • Line 613: “… biofilm …” should be “… biofilms …”

All the minor corrections are made in the manuscript according to the reviewer suggestions.

Reviewer 2 Report

The revised manuscript is significabtly better than the original. After my opinion, this is a valuable contribution to the field and a manuscipt acceptable for publication. 

Author Response

We would like to thanks reviewers for critically reviewing the manuscript. 

This manuscript is a resubmission of an earlier submission. The following is a list of the peer review reports and author responses from that submission.

Round 1

Reviewer 1 Report

The review manuscript of Yadav et al is an interesting piece of work and collates important data on the building blocks of biofilm formation in Streptococcal species and also briefly outlies potential and current therapeutic strategies to combat related biofilm growth. The review brings together past and recent research and gives an overview of molecules used by Streptococcal species in biofilms. The main focus of the article is a catalogue of proteinaceous biofilm molecules, which is very useful but at times could be better laid out (e.g. in places feels like a long list of names but no real mechanistic appreciation of how they mediate biofilm growth). This is the main limitation of the review: a little too brief at times. In places this needs a bit of work needs to be done to rectify this. In general, I think the review is interesting but there are shortcomings that should be addressed to improve the manuscript.

  • The review is lacking figures. It is therefore hard to follow at time (lots of text). One figure which is needed is a schematic showing the general flow of biofilm formation in Streptococci highlighting where different types of molecules are most abundant. These details should also be included in the text (section 1.0). For example, pili/adhesins predominate in early formation; amyloid generally essential for early and maturation; eDNA released during maturation; carbohydrates major EPS component throughout.

  • Section 2.1: Please provide more information on the carbohydrates of GAS – e.g. how do L-glucose and D-mannose form glycan chains – what is their linkage etc.

  • Line 160: give details of which pili proteins have been documented

  • Section 2.5: when describing “glucans” give details of which type they are. These can be diverse with different functions so understanding the differences is useful here

  • Section 2.7: It would be useful to expand on how glucans are synthetised from sucrose. It is not clear as it is currently written. GTF proteins are named incorrectly – it should be GtfB, GtfC, GtfD etc. this should be checked for other protein names throughout.

  • Section 3.1: A large part of the review is focussed on surface proteins and pili. A figure is needed to show a schematic of the general pilus architecture. Also, in this section there is a large discussion of different pili from different bacteria, but only a few have been placed in tables (Table 1+2). What is the rational for this? If there is none, the authors should list all pili and components. Also, why are only 7 pili described in table 1, while there are 9 different types?

  • Section 3 in general: there is an extensive list of different proteins but no real discussion of how they are mediating biofilm formation. A more detailed explanation of their role would be useful – which mediate cell adhesion, which bind abiotic surfaces; which promote bacterial aggregation; and is there and understanding of how they do this (e.g. which host receptors, do pili interact with one another etc).

  • Table 1 legend: “different FCT types”? types of what? Pili?

  • Section 3.2: This section is very busy and a bit confusing. It seems to contain all other proteins on the surface, even though they have very different roles. The section should be divided into “Adhesins” and “Other surface proteins”. There is also a jumping around where proteins are repeated later. Later AgI/II is mentioned but not in relation to it being able to forma amyloid. Baps are then later discussed, which also form amyloid, but they are seemingly linked with Fap1, which does not form amyloids. Fap1 is a serine-rich fimbriae but these class of proteins are not described anywhere, although they are important across This whole section needs to be made clearer with better flow and connections between the molecules being discussed.

  • Tables 3 to 7: I do not really understand these tables. They contain lots of proteins that are not mentioned in the text. They should either be discussed or removed. The tables should also be combined and made more concise.

  • Figure 1A: the ellipsoid should be removed as it overlaps with the text making it hard to read.

  • Figure 1B: the legend needs expanding on

  • There needs to be a conclusion to bring the review together at the end and it would be useful for the authors to give their thoughts on future directions in the field

I think in general the manuscript is well written but I have identified a couple of minor mistakes:

  • Line 413: QS is mentioned for the first time – please explain the acronym on first usage.
  • Line 452: ciaH is a gene and should be in italics

Reviewer 2 Report

This is a well written and comprehensive review article that I really appreciated to read. I have only a few comments (minor):

  1. The title has a potential to be better. As it stands now it seems like the strep biofilms can be used for therapeutic strategies. Rephrasing the title will improve the first impression of the article. 
  2. The introduction that is not strep specific is a bit long. Consider to shorten this section or remove part 1.1 to 1.3 as these are discussed in more detail for streptococci later in the manuscript.